# Elution Kinetics from Antibiotic-Loaded Calcium Sulfate Beads, Antibiotic-Loaded Polymethacrylate Spacers, and a Powdered Antibiotic Bolus for Surgical Site Infections in a Novel In Vitro Draining Knee Model

**DOI:** 10.3390/antibiotics10030270

**Published:** 2021-03-08

**Authors:** Kelly Moore, Rebecca Wilson-van Os, Devendra H. Dusane, Jacob R. Brooks, Craig Delury, Sean S. Aiken, Phillip A. Laycock, Anne C. Sullivan, Jeffrey F. Granger, Matthew V. Dipane, Edward J. McPherson, Paul Stoodley

**Affiliations:** 1Wexner Medical Center, Department of Microbial Infection and Immunity, The Ohio State University, Columbus, OH 43210, USA; Kelly.moore@osumc.edu (K.M.); Devendra.dusane@nationwidechildrens.org (D.H.D.); brooks.922@buckeyemail.osu.edu (J.R.B.); 2Biocomposites Ltd., Keele Science Park, Keele, Staffordshire ST5 5NL, UK; rkw@biocomposites.com (R.W.-v.O.); cpd@biocomposites.com (C.D.); sa@biocomposites.com (S.S.A.); pl@biocomposites.com (P.A.L.); 3Center for Clinical and Translational Research, The Research Institute at Nationwide Children’s Hospital, Columbus, OH 43210, USA; jgranger230@gmail.com; 4Wexner Medical Center, Department of Orthopaedics, The Ohio State University, Columbus, OH 43210, USA; anne.sullivan@osumc.edu; 5Department of Orthopaedic Surgery, David Geffen School of Medicine at UCLA, Santa Monica, CA 90403, USA; mdipane@mednet.ucla.edu (M.V.D.); emcpherson@mednet.ucla.edu (E.J.M.); 6National Centre for Advanced Tribology at Southampton (nCATS), National Biofilm Innovation Centre (NBIC), Department of Mechanical Engineering, University of Southampton, Southampton SO17 1BJ, UK

**Keywords:** periprosthetic joint infection, antibiotic loaded calcium sulfate beads, polymethylmethacrylate (PMMA) bone cement, vancomycin, tobramycin

## Abstract

Antibiotic-tolerant bacterial biofilms are notorious in causing PJI. Antibiotic loaded calcium sulfate bead (CSB) bone void fillers and PMMA cement and powdered vancomycin (VP) have been used to achieve high local antibiotic concentrations; however, the effect of drainage on concentration is poorly understood. We designed an in vitro flow reactor which provides post-surgical drainage rates after knee revision surgery to determine antibiotic concentration profiles. Tobramycin and vancomycin concentrations were determined using LCMS, zones of inhibition confirmed potency and the area under the concentration–time curve (AUC) at various time points was used to compare applications. Concentrations of antibiotcs from the PMMA and CSB initially increased then decreased before increasing after 2 to 3 h, correlating with decreased drainage, demonstrating that concentration was controlled by both release and flow rates. VP achieved the greatest AUC after 2 h, but rapidly dropped below inhibitory levels. CSB combined with PMMA achieved the greatest AUC after 2 h. The combination of PMMA and CSB may present an effective combination for killing biofilm bacteria; however, cytotoxicity and appropriate antibiotic stewardship should be considered. The model may be useful in comparing antibiotic concentration profiles when varying fluid exchange is important. However, further studies are required to assess its utility for predicting clinical efficacy.

## 1. Introduction

Periprosthetic joint infections (PJI) present many challenges for orthopedic surgeons as the economic burden and incidence rate increases [1]. These infections lengthen hospitalizations and increase the risk of mortality and morbidity [2]. Antibiotic tolerant bacterial biofilms play a major role in the pathogenesis and treatment of (PJI) [3]. Due to a higher tolerance to antibiotics compared with planktonic cells, biofilms require extended or repeated high concentrations of exposure [4,5].

To treat bacterial biofilm PJIs, antibiotics such as vancomycin and the aminoglycosides tobramycin and gentamicin are often mixed into bone cement to provide locally high concentrations and broad spectrum coverage against Gram positive and negative pathogens [6]. Broad spectrum antibiotics are used because culture data may not identify the infecting bacteria or this information may not be available at the time of operation [7,8]. Systemic antibiotic administration can decrease overall infection rates, however there is evidence that local administration further reduces infection [9]. Historically, surgeons have used antibiotic loaded polymethylmethacrylate (PMMA) bone cement in the form of spacers and beads as methods of local administration [4]. Antibiotic loaded calcium sulfate (CaSO_4_) hemihydrate void filler beads have also been used because they are compatible with many classes of antibiotics [10] and because they are fully absorbable promoting the release of all antibiotic [11,12,13]. More recently, vancomycin powder has been directly administered intraoperatively in revision hip and knee arthroplasty [14].

Elution kinetics of antibiotics within each body cavity is an important factor in treating bacterial biofilm infections. It is important for the antibiotics to maintain or surpass the minimum inhibitory concentration (MIC), as bacterial biofilms require higher than MIC concentrations for an extended time to eradicate biofilm bacteria [15]. Whilst the concentrations of antibiotics are crucial to treatment and prevention of infection, it is important to know whether the antibiotics are released in bursts, at a steady rate, or if they are washed out by fluid exchange in the body. Conventional laboratory elution kinetic studies use closed systems where the antibiotic containing depot is placed into a known volume of liquid and periodically completely, or partially, exchanged, and the concentration in which the antibiotics are released overtime is measured [16,17,18]. This method provides a quantitative mathematical description of the elution kinetics [19], but neglects to consider continuous fluid exchange which changes in the joint space as observed from clinically measured draining rates [20,21,22,23]. Further, such methods cannot be used to assess the comparative pharmacodynamics of a powder antibiotic bolus over time since all of the antibiotic is released instantaneously, making the cumulative release instantly equal to the final concentration. 

The aim of the present study was to develop a method to allow direct comparison of the antibiotic concentration profile when released from non-absorbable and absorbable depots compared to direct addition of antibiotic as a powder in an artificial draining knee model. We designed a continuous flow reactor system to mimic drainage rates following revision which was in the size range of a human knee, to support a full sized PMMA bone cement spacer as well as a clinically relevant number of absorbable calcium sulfate beads. Using this system, we measured the antibiotic profile kinetics of the three local antibiotic administration methods using Liquid Chromatography Mass Spectrometry (LCMS) and the potency of the antibiotic was confirmed against reference strains of *Staphylococcus aureus* and *Pseudomonas aeruginosa* using a diffusion filter disc assay. We hypothesized that the addition of antibiotic loaded beads to an antibiotic loaded PMMA spacer would produce a greater area under the curve of antibiotics over 48 h than the PMMA spacer alone and the vancomycin powder would be rapidly washed out. Our primary outcome measure was the area under the curve (AUC) of antibiotic concentration at various timepoint over a period of 48 h. 

## 2. Results

### 2.1. Drainage Flow

Previous studies by Fan, Boland and Porteous observed drainage rates after knee arthroplasty revision surgeries [21,22,23]. Using data reported from their studies, a trendline of y = 3.1269x^−1.019^, where y = flow rate and x = time, was calculated and used to determine the flow rates used in this study (Figure 1). Since we could not operate the pump using a continuous function, we imposed a stepwise decrease to match the curve. The initial flow of 3.5 mL/min was visually back extrapolated from the clinical data. 

### 2.2. Calcium Sulfate Beads Stability

After 48 h, calcium sulfate beads were visible within the system. Once removed, the beads had appeared to have softened compared to when they were introduced to the system, but were still structurally sound (Figure 2). 

### 2.3. Liquid Chromatography Mass Spectrometry (LCMS)

Data were reported on a Log2 scale to show potency and concentrations at time points where samples were taken frequently and drainage rates were rapidly changing. Antibiotic concentration quantified using LCMS is presented as the geometric mean of the Log10 transformed concentration with SE bars for vancomycin (VAN) (Figure 3A) and tobramycin (TOB) (Figure 4A).

While VP had the greatest release of VAN for the first 30 min of operation (Figure 3A), after 2 h, these concentrations were below the minimum detection limit of 0.98µg/mL (−0.01 ug/mL expressed as a base Log_10_). The CSB_v+t_plusSpacer_v+t_ released greater VAN concentrations than the Spacer_v+t_ throughout the duration of the study, including up to a 2.3 log difference at 48 h (Figure 3B) (*p* = 0.0024) and were significantly different at 5 min, 3 h and every subsequent timepoint until the study was completed at 48 h (*p* < 0.05 at all timepoints in this period).

The CSB_v+t_plusSpacer_v+t_ and Spacer_v+t_ showed a similar trend in terms of detectable concentrations of TOB (Figure 4A). The CSB_v+t_plusSpacer_v+t_ showed a trend of greater TOB concentrations than the Spacer_v+t_ alone throughout the duration of the study, by as much as an 0.77 log difference at h 5 and 6 (Figure 4B). This difference was statistically significant between 2 and 5 h (*p* < 0.05 at all timepoints).

### 2.4. Antibiotic Profile Measured as Area under the Curve (AUC)

The CSB_v+t_plusSpacer_v+t_ achieved greater antibiotic AUC of TOB at all time points and was significantly greater at 12 and 24 h compared to the Spacer_v+t_ alone (*p* = 0.00073 and 0.0071, respectively) (Figure 5). For VAN, the greatest AUC at 2 h was by VP but the difference was not statistically significant (*p* < 0.05). However, there was a steady increase in AUC achieved by the Spacer_v+t_ and the CSB_v+t_plusSpacer_v+t_ which became greater (but not statistically *p* > 0.05) than that of the VP at 12 h. The rate of increase in AUC achieved by the CSB_v+t_plusSpacer_v+t_ was greater than the Spacer_v+t_ alone which was significantly greater (*p* < 0.05) at all time points. 

### 2.5. Zone of Inhibition (ZOI)

Generally, the ZOI data were complementary to the LCMS data. Data were reported on a Log2 scale to clearly show potency at time points where samples were taken frequently and drainage rates were rapidly changing. VAN and TOB potency against *S. aureus* UAMS-1 (Figure 6A) and *P. aeruginosa* PAO1 (Figure 6B), respectively was quantified as ZOI and reported as mean with SE bars. 

Against *S. aureus* UAMS-1, VP effluent had the greatest potency compared to the Spacer_v+t_ (*p* < 0.05 at all timepoints) and the CSB_v+t_plusSpacer_v+t_ for the first 20 min of operation. However, after 2 h of initiating flow rates, VP lost all potency (Figure 6A). Similarly, the Spacer_v+t_ effluent showed potency for the first 30 min, then lost all potency after 2 h, however, potency was regained at 12 h as evidenced by a ZOI. The CSB_v+t_plusSpacer_v+t_ maintained a relatively uniform potency creating a consistently greater ZOI than effluent from the Spacer_v+t_ alone and was statistically greater from the 40 min timepoint until the 24 h timepoint (*p* < 0.05 at all timepoints in this period).

Against *P. aeruginosa* PAO1, potency patterns demonstrated a similar trend to *S. aureus* (Figure 6B). The Spacer_v+t_ demonstrated potency for the first 45 min before losing all potency, and subsequently regained potency at 24 h. Initially, the CSB_v+t_plusSpacer_v+t_ demonstrated high potency levels which slowly decreased over the course of 2 h. After 2 h, that degree of potency was relatively constant for the remainder of the study. The CSB_v+t_plusSpacer_v+t_ effluent generated a statistically greater ZOI than the Spacer_v+t_ alone for all timepoints except at 20 min (*p* < 0.05 at all other timepoints). 

## 3. Discussion

In our study, we introduce a continuous flow-through reactor system designed based on drainage in the human knee following PJI revision surgery, to study the antibiotic concentration profile over time based on the elution from antibiotic loaded PMMA bone cement, absorbable CSB and as a single bolus added in powder form. The continual replenishment of eluant creates an “infinite sink” for the antibiotic [24]. Flow through reactor systems have been used to study the desorption of chemicals from soils as well as elution of antibiotics from bone cements [25]. Infinite sink conditions are important since the diffusion flux of the component from the solid phase surface (in our case the antibiotic depot) into the surrounding eluant is a function of the concentration gradient and thus by matching, as best we could, physiologically relevant volumes and drainage rates we attempted to recreate these conditions in an in vitro model. An infinite sink flow through reactor system also has the advantage that the concentration profile of an antibiotic added as a powder directly to the reactor can be measured.

The fluctuating concentration profiles demonstrated the utility of the flow through system with variable flow. The vancomycin powder initially demonstrated high concentrations in the effluent but was steadily and rapidly washed out so that after two hrs it was below detection limits by LCMS (Figure 3A) and failed to produce a zone of inhibition (Figure 6A). Recently, intraarticular placement of vancomycin powder has been used in primary knee and hip arthroplasty [26]. Otte et al. reported intrawound placement of vancomycin reduced infection [14] however other studies have shown that vancomycin powder had no effect on preventing infection [27,28]. It is possible that this lack of efficacy may be due to rapid washout of vancomycin, as would be predicted from our study. Our model is intended to compare antibiotic release kinetics when loaded into bone cement and absorbable depots or as applied in powder form in the context of infected arthroplasties where biofilm is suspected. Pharmacodynamic data on vancomycin administered as a powder in total joint arthroplasty are sparse, but Johnson et al. estimated a half-life of 7.2 h [29]. In our study we saw a linear decrease rather than an exponential decrease, which was likely due to incomplete mixing in the reactor and changing of the flow through flow rate, our “half-life” was much lower at approximately 15 min. A number of factors might account for such a large discrepancy. The Johnson et al. study did not report flow rates and it is possible that their flow rates might have been less than those from the clinical values we used, which would be particularly important immediately post operatively when drainage rates we used were high. Second, in the human knee it is likely there are “dead zones” with no, or little liquid flow from which the vancomycin powder could slowly diffuse. Lastly, surrounding soft tissue and bone may absorb vancomycin, which is subsequently released by desorption or elution over time. However, concentration-time data from studies such as the Johnson et al. study will be useful for us to refine flow rates, volume, and degree of mixing in our model to more closely match clinical observations. 

The concentration kinetic profile of antibiotics released from PMMA and CSB was different from that of the powder bolus. There was an initial high concentration, which presumably occurred from burst release in the period between adding the eluant and the time taken to collect the first sample. There was then a steady decrease in concentration for the first hr or two before the concentration increased again. The increase can be contributed to the reduction in flow through flow rate, so while elution likely remained relatively constant over this time, the washout rate was reduced since the concentration in the system is equal to the rate of elution minus the rate of washout, assuming antibiotic is not being degraded. Such fluctuations would not be seen in release assays with periodic exchanges, however further clinical data or appropriate animal PJI models are required to demonstrate how well our model compares to antibiotic kinetics delivered locally in a human joint. While these patterns were similar for antibiotic loaded PMMA with and without CSB, the concentration of both vancomycin and tobramycin was higher when both were combined. A previous study used partial fluid exchange and found the addition of vancomycin loaded CSB to PMMA spacers increased concentrations of antibiotics administered [30]. We found similar results in our model as CSB increased the vancomycin concentration up to 2.28 log compared to the PMMA alone (Figure 3B) (*p* = 0.0024). For tobramycin, the addition of CSB increased concentrations by a log difference of 0.77 (Figure 4B) (*p* = 0.014). The ZOI assays were generally consistent with the LCMS data showing that the antibiotics did retain potency after elution. 

Our AUC data showed how the antibiotic exposure varied over time using these different administration methods. For vancomycin, the powder bolus had the greatest area under the curve (AUC) at 2 h, however at later time points the CSB_v+t_plusSpacer_v+t_ became greatest as antibiotic continued to elute (Figure 5). For both vancomycin and tobramycin the AUC achieved by the CSB_v+t_plusSpacer_v+t_ was greater than by the Spacer_v+t_ alone at all time points. This was significantly greater at all timepoints for vancomycin (*p* < 0.05 at all time points) and significantly greater at h 12 and 24 for tobramycin (*p* = 0.00073 and 0.0071, respectively). Badha et al. reported the minimal biofilm eradication concentration (MBEC) on bone and muscle to be 100–750 ug/mL of a 1:1 vancomycin and tobramycin combination maintained for a minimum 24 h [31]. Our model indicates concentrations of this magnitude were achieved with VP and the CSB_v+t_plusSpacer_v+t_, but only the CSB_v+t_plusSpacer_v+t_ maintained levels above this concentration for the 24 h duration (Figure 3A and Figure 4A). 

We acknowledge several limitations of this study. First, only one commercially available calcium sulfate material was used. Not all commercially available materials have the same ability to mix and may not have the same elution profile [32]. Second, our model has no soft tissue mimic to simulate absorption and later release. Finally, although we saw consistent trends in our replicate runs, we note variability in our data. The mixing kinetics in the reactor are not well characterized and the presence of the PMMA and beads will almost certainly ensure there is not complete mixing, this could result in channeling and “dead” zones which could change as the beads and spacer shift during rocking. The second likely source of variation is in difficulty in synchronizing timing in taking the samples and manually adjusting flow rates from run to run. This would be particularly problematic in the early time points when sampling and flow rate changes are most frequent. Automated pump operation and sampling may reduce this variability. Nevertheless, we believe our model may have utility for directly comparing antibiotic concentration profiles in situations where drainage or inflammation fluids may dilute and wash out locally administered antibiotics.

In conclusion VP initially resulted in the largest vancomycin concentrations, however, within 2 h, concentrations dropped below inhibitory levels. CSB_v+t_plusSpacer_v+t_ provided higher concentrations of antibiotics and a larger AUC at 24 and 48 h than other methods assessed. This suggests the combination of PMMA and CSB may present an optimal combination for killing biofilm bacteria; however, cytotoxicity and appropriate antibiotic stewardship should be considered. We further conclude that the model may be useful in comparing the antibiotic concentration profiles from different local applications when varying fluid exchange is important but recognized that the model is relatively complex to operate and that further in vitro and animal studies are required to assess its utility for predicting clinical efficacy.

## 4. Materials and Methods

### 4.1. Reactor System

A continuous flow reactor system (Figure 7) was designed using a 500 mL beaker, the lid of a vacuum filter unit (“Nalgene” Rapid Flow Sterile Single Use, Thermo Scientific), LS-14 tubing (Platinum-Cured Silicone, Precision Pump Tubing, Masterflex) and an IPC High Precision Multichannel Pump (ISMATEC, Cole-Parmer, Vernon Hills, IL, USA). The reactor vessel consisted of the beaker, 75 mL of full-strength Ringers solution (Sigma-Aldrich, St. Louis, MO, USA) and an antibiotic administration method. Holes were created in the lid of the vacuum filter unit for the tubing to fit through and was secured to the top of the beaker. The pump was used to transfer full strength Ringers solution into and out of the reactor vessel simulating clinical values of post-surgical drainage after knee arthroplasty revision surgery (Figure 1) [21,22,23]. The reactor vessel was placed on a laboratory platform rocker (VSR-50 PRO Scientific, Oxford, CT, USA) at a frequency of 5 RPM to create fluid motion minimizing dead-end zones in the system and to mimic the mechanics of fluid within the knee joint [33]. The 75 mL of full-strength Ringers solution was first added to the reactor vessel before antibiotic administration methods were applied. The pump was then activated to pump Ringers solution in and out of the reactor vessel at equal rates to maintain a constant volume of 75 mL. 

Once drainage flow was initiated, 2.5 mL of effluent samples were manually collected at 5, 10, 15, 20, 25, 30, 35, 40, 45, 50, and 55 min, and 1, 2, 3, 4, 5, 6, 7, 8, 12, 18, 24, and 48 h thereafter. The samples were stored at −20 °C until ready for analysis when the samples were defrosted on ice to room temperature.

### 4.2. Preparation of Antibiotic Vehicles

For our antibiotic choice we used vancomycin (VAN) and tobramycin (TOB) as these antibiotics are commonly mixed into bone cement and absorbable beads in revision surgery to provide locally high concentrations over sufficient time to control biofilm bacteria [6,34,35]. 

#### 4.2.1. Preparation of Antibiotic Loaded Calcium Sulfate Beads 

Antibiotic loaded calcium sulfate beads (CSB_v+t_) of size 4.8 mm were prepared using Stimulan^®^ Rapid Cure (SRC) (Biocomposites Ltd., Staffordshire, UK) 10-cc mixing kits. Then, 20 g of CaSO_4_ SRC powder was combined with 240 mg Tobramycin Sulfate (TOB) (Gold Biotechnology Inc., Olivette, MO, USA) and 1000 mg Vancomycin Hydrochloride (VAN) (Gold Biotechnology Inc., Olivette, MO, USA) [36]. After combining the dry components, 6 mL of the liquid mixing solution, included in the kit, was added. All components were stirred for 30 s until the materials came together as a paste, which was transferred and pressed into a mold mat (Biocomposites Ltd., Staffordshire, UK). The paste was set for 10–15 min at 20 °C before being removed as solid beads and added to the reactor system. 

#### 4.2.2. Preparation of Antibiotic Loaded and Unloaded Polymethylmethacrylate (PMMA) Bone Cement Space Mimics

PMMA bone cement spacer mimics were prepared using Simplex^™^ P SpeedSet^™^ Radiopaque Bone Cement construction kits (Stryker^®^ Howmedica Osteonics, NJ, Mahwah, USA). Each unloaded spacer (Spacer_u_) was prepared combining 40 g of Simplex^™^ PMMA bone cement powder and 20 mL of the methyl methacrylate liquid monomer included in the kit. The contents were mixed into a uniform paste that was added to a circular silicone mold mat of 3.45 cm in diameter (Silikomart Professional Silicone Baking Mold, Cylinder 6 Cavities, Amazon, WA, USA). The spacers were removed after approximately 30 min at 20 °C [37]. The antibiotic loaded spacer (Spacer_v+t_) was prepared using the same protocol, but included adding 2 g VAN and 2 g TOB to the dry mixture before adding the liquid monomer similar to the CSB_v+t_ procedure above. 

#### 4.2.3. Preparation of Vancomycin Powder Bolus (VP) 

A VAN bolus of 1000 mg VAN powder (VP) was weighed out and directly poured into the reactor system immediately before initiating flow. 

### 4.3. Methods of Antibiotic Administration

Three different methods of antibiotic administration were studied: (a) an antibiotic-loaded spacer alone, (b) an antibiotic-loaded spacer and antibiotic-loaded CSB, and (c) an unloaded spacer and vancomycin powder bolus (VP). For the latter, the unloaded spacer was used to ensure similarity in volume and liquid levels between the three experimental arms. Each arm was performed in three independent replicates.

### 4.4. Liquid Chromatography Mass Spectrometry (LCMS)

Quantification of antibiotics was developed using a single quadrupole mass spectrometer coupled to a 1260 infinity II series liquid chromatography stack (LCMS) (Agilent Technologies. Inc, Santa Clara, CA, USA). Liquid chromatography separation was conducted with a Poroshell 120-SB-C18 column (2.1 × 100 mm, 2.17 µm) (Agilent Technologies, Inc, Santa Clara, CA, USA). Determination of vancomycin and tobramycin concentrations were assessed based on an adaptation of a previously published method [38].

### 4.5. Area under the Curve (AUC)

In order to test our primary outcome and compare the antibiotic exposure provided by each administration method Prism 8 (Graph Pad Version 8.4.3) was used to determine the area under the curve (AUC) of antibiotic concentrations from the linear data. Using this software, the AUC was determined for the 2 h, 12 h, 24 h and 48 h time points for each antibiotic administration method and for both VAN and TOB concentrations.

### 4.6. Kirby-Bauer Assessment of Effluent Potency by Zones of Inhibition (ZOI)

To assess the VAN and TOB potency of each effluent sample, the zone of inhibition (ZOI) was measured using the Kirby-Bauer method [39]. While this method is often used to test susceptibility, here we use the assay to semi-quantitatively measure the potency and concentration of antibiotic to corroborate the LCMS data. Two reference strains, *Pseudomonas aeruginosa* PAO1 (MIC_TOB_ 1.5 µg/mL, MIC_VAN_ > 64 µg/mL) and *Staphylococcus aureus* UAMS-1 (MIC_TOB_ 2 µg/mL, MIC_VAN_ 2 µg/mL) were used in this study to determine the antimicrobial potency of each effluent sample taken at different time points. *P. aeruginosa* and *S. aureus* were grown overnight in Tryptic Soy Broth (TSB; Becton, Dickinson & Company, Sparks, MD, USA) at 37 ℃ under shaker conditions (200 RPM). Each overnight culture was diluted 1%. Then, 100 µL of the diluted culture was added to a sterile Petri dish containing 24 mL of 1.5% Tryptic Soy Agar (TSA; Sigma-Aldrich, St. Louis, MO, USA) and spread to cover the surface. Each plate was left to dry for approximately 10 min before adding six sterile filter discs (7 mm, Whatman GF/F Glass microfiber, Sigma Aldrich, St. Louis, MO, USA) in a circular formation with one disc in the center. Then, 10 µL of each effluent sample was pipetted onto its corresponding disc then incubated for 24 h at 37 ℃. After incubation, an image was taken of each Petri plate to analyze the zone of inhibition (ZOI) using NIH ImageJ [40]. ImageJ image analysis software was used to measure the diameter of each effluent sample ZOI. Each image was calibrated to the 100 mm Petri dish to convert pixels to cm.

### 4.7. Statistical Analysis

Each antibiotic administration arm was performed in independent triplicate runs and the average of each method was used for analysis. The LCMS data and AUC data were converted to the geometric mean by taking the Log of each value before calculating the average values used for analysis. Statistical comparisons between different antibiotic administration methods were completed through Excel (Microsoft Version 16.39) software using an unpaired two-tailed, Student’s t-test assuming equal variances. A P-value less than 0.05 was considered statistically significant.

Log differences between antibiotic administration methods were completed for LCMS data through Excel (Microsoft Version 16.39). The log of each value was calculated before subtracting Spacer_v+t_ values from CSB_v+t_plusSpacer_v+t_ and then plotted.

## Figures and Tables

**Figure 1 antibiotics-10-00270-f001:**
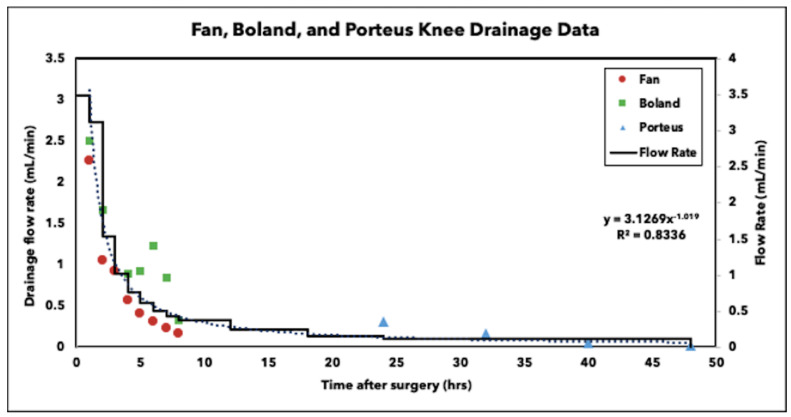
Drainage flow observed after knee arthroplasty revision surgery from Fan, Boland and Porteous studies. The curve fit data were used to generate the stepwise flow rate decrease in the reactor system.

**Figure 2 antibiotics-10-00270-f002:**
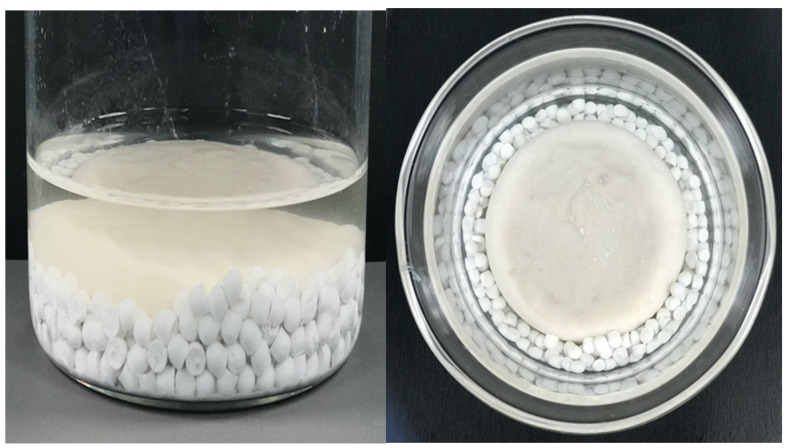
Representative images of calcium sulfate beads within the reactor system after 48 h.

**Figure 3 antibiotics-10-00270-f003:**
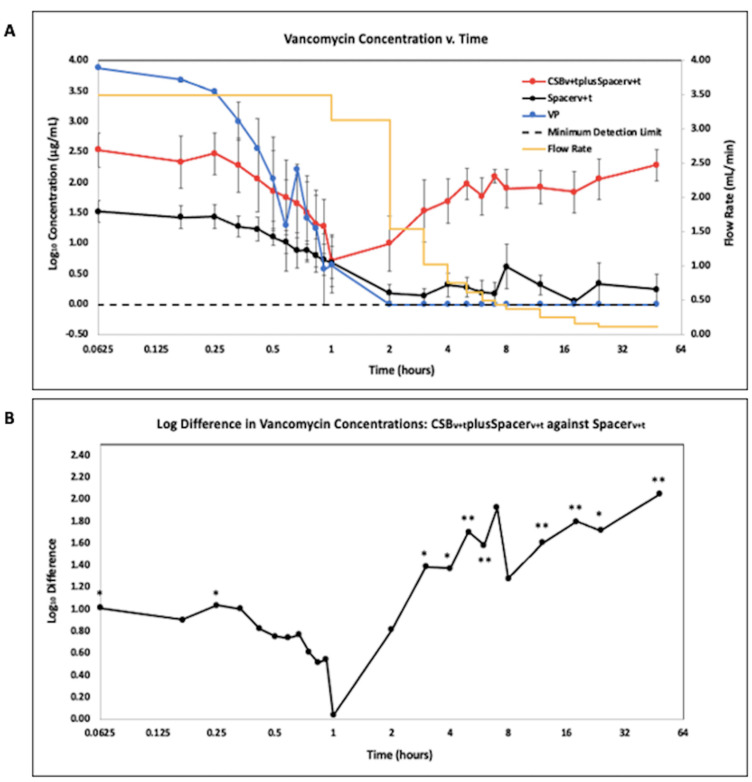
(**A**). LCMS analysis of vancomycin in effluent samples from antibiotic loaded calcium sulfate beads with an antibiotic loaded PMMA spacer (CSB_v+t_plusSpacer_v+t_), antibiotic loaded PMMA spacer (Spacer_v+t_), and vancomycin powder bolus (VP). N = 9; geometric mean ±SE. The yellow curve is the influent saline flow rate. Time is shown on a log2 scale to more clearly show the changes in concentration at the early time points. (**B**). Log difference of vancomycin effluent concentrations of the CSB_v+t_plusSpacer_v+t_ compared to the Spacer_v+t_ alone. (* *p* < 0.05, ** *p* < 0.01).

**Figure 4 antibiotics-10-00270-f004:**
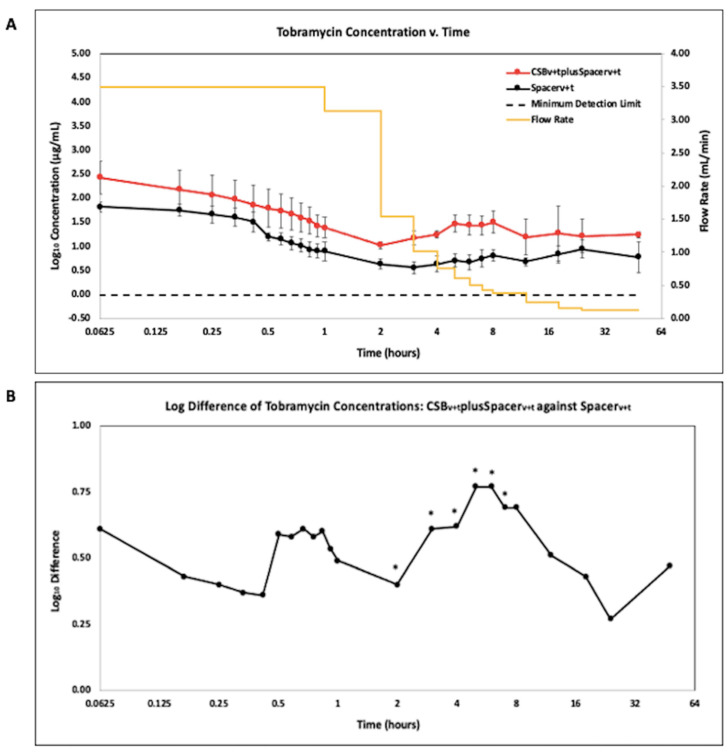
(**A**). LCMS analysis of tobramycin in effluent samples from antibiotic loaded calcium sulfate beads with an antibiotic loaded PMMA spacer (CSB_v+t_plusSpacer_v+t_), and antibiotic loaded PMMA spacer (Spacer_v+t_). N = 6; geometric mean ± SE. The yellow curve is the influent saline flow rate. Time is shown on a log2 scale to show the changes more clearly in concentration at the early time points. (**B**). Log difference of tobramycin concentrations of the CSB_v+t_plusSpacer_v+t_ method compared to the Spacer_v+t_ method. (* *p* < 0.05).

**Figure 5 antibiotics-10-00270-f005:**
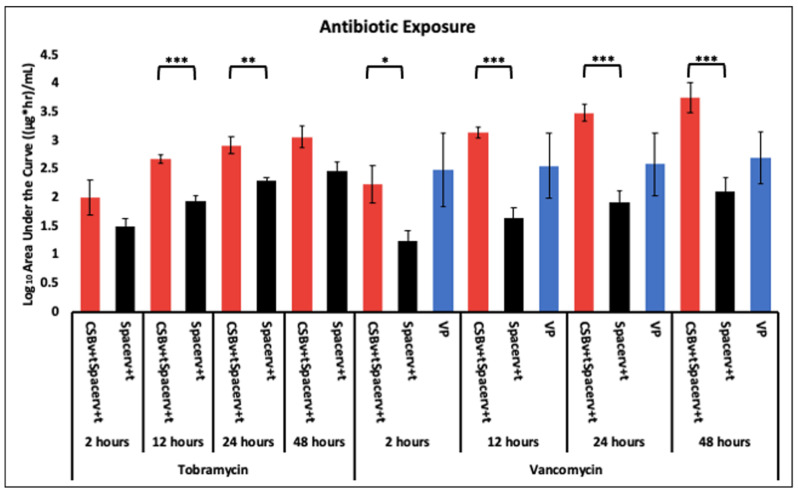
Tobramycin and vancomycin exposure via antibiotic administration methods to the system at 2, 12, 24 and 48 h. N = 4 replicates for Spacerv+t and CSBv+tplusSpacerv+t and 3 for VP; geometric mean ± 1 SE. (* *p* < 0.05, ** *p* < 0.01, *** *p* < 0.001). The same data shown on a linear Y-axis scale are available in Appendix A.

**Figure 6 antibiotics-10-00270-f006:**
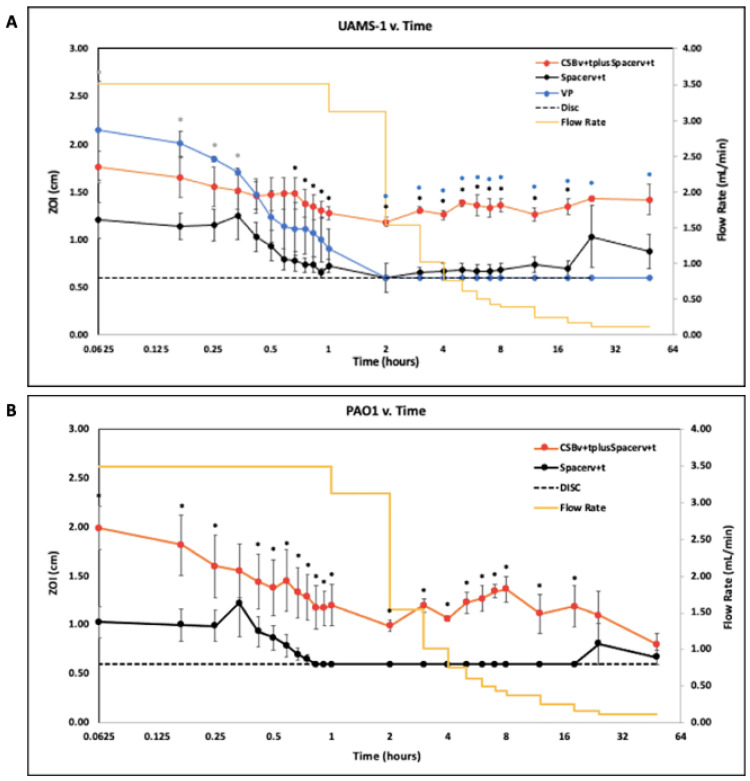
(**A**). Potency of drainage effluent from the reactor measured as the ZOI of effluent against *S. aureus* UAMS-1 reported on a Log2 scale for 48 h after flow rates were initiated. The dashed line indicates the diameter of the disc. When the ZOI of the effluent was equivalent to the diameter of the disc, it indicates growth up to the edge of the disc and is assumed there was no antimicrobial potency. (* *p* < 0.05 for the CSB_v+t_plusSpacer_v+t_ compared to the Spacer _v+t_ alone, * *p* < 0.05 for the CSB_v+t_plusSpacer_v+t_ compared to the VP, and * *p* < 0.05 for the Spacer_v+t_ alone compared to the VP) N = 3, mean ± SE. (**B**). ZOI of effluent against *P. aeruginosa* PAO1 reported on a Log2 scale for 48 h after flow rates were initiated. The dashed line indicates the diameter of the disc. N = 3, mean ± SE. (* *p* < 0.05 for the CSB_v+t_plusSpacer_v+t_ compared to the Spacer _v+t_ alone).

**Figure 7 antibiotics-10-00270-f007:**
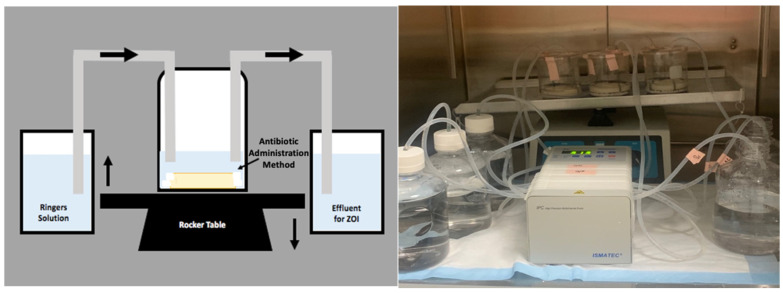
Schematic and real image of the in vitro draining knee reactor system. The system was placed onto a VSR-50 laboratory platform rocker at a frequency of 5 RPM. Ringers solution was pumped into and out of the reactor vessel at flow rates derived from post-operative surgical drainage rates.

## Data Availability

The data presented in this study are available in Appendix A.

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
