# Peer review of "Elution Kinetics from Antibiotic-Loaded Calcium Sulfate Beads, Antibiotic-Loaded Polymethacrylate Spacers, and a Powdered Antibiotic Bolus for Surgical Site Infections in a Novel In Vitro Draining Knee Model"

_antibiotics, 2021, doi:10.3390/antibiotics10030270_

Round 1
Reviewer 1 Report
This a very well written and useful paper. This model can be extrapolated for other clinical situations, in which the previous knowledge of the antibiotic release curve is critical for the course of infection.
However, I think that the Kirby-Bauer Assessment of Effluent Potency By Zones of Inhibition must be better clarified. As we all know, the Kirby Bauer method is standardized for antibiotic disks containing a certain antibiotic concentration. Thus, I am not sure that the measurement of ZOI produced by variable concentrations of antibiotics contained in the effluent could be extrapolated in active concentrations. Maybe an MIC assay would have been more appropriate.
Author Response
This a very well written and useful paper. This model can be extrapolated for other clinical situations, in which the previous knowledge of the antibiotic release curve is critical for the course of infection.
However, I think that the Kirby-Bauer Assessment of Effluent Potency By Zones of Inhibition must be better clarified. As we all know, the Kirby Bauer method is standardized for antibiotic disks containing a certain antibiotic concentration. Thus, I am not sure that the measurement of ZOI produced by variable concentrations of antibiotics contained in the effluent could be extrapolated in active concentrations. Maybe an MIC assay would have been more appropriate.
Response: We agree that the Kirby-Bauer assay has a specific use for assessing antibiotic susceptibility. However, here we use the assay not to demonstrate susceptibility (we have now reported the MICs of each strain against the antibiotic) but use the zone of inhibition (ZOI) as a semi-quantitative measure of potency with which to corroborate the LCMS data. We have now made this point clear in the text. We also refer to the assay as a Kirby-Bauer-like disc diffusion assay. Page 11 lines 1049-1051.
Reviewer 2 Report
The authors develop a method to allow direct comparison of the antibiotic concentration profile from non-absorbable, absorbable and powder depots in an artificial draining knee model. They designed a continuous flow reactor system that was programed to mimic drainage rates following revision and was similar in size to a knee- supporting a human sized PMMA bone cement spacer, absorbable calcium sulfate beads and antibiotic powder administered as a bolus. The reactor system mimicked post-surgical drainage following knee arthroplasty revision surgery to account for fluid exchange. They concluded that the combination of a vancomycin bolus, PMMA and CSB may present an optimal combination for killing bacteria entering the surgical site and provide long term protection against subsequent biofilm formation.
Major objections:
- Although the manuscript is well written it is not easy to follow. There is too much text, some parts could be more easily structured. Introduction should be shortened. Results should be structured in a Table.
- From surgical point of view the most important question is the possibility of the clinical implication of the examined system. The conditions created by the authors in vitro are not realistic and are questionable in real life. In their model there is no soft tissue, there is no real absorption and later release. We do not know which amount of antibiotic should be absorbed by the tissues. Also there is no immune response. All of that should play a role in vivo. I am concerned whether there is any possibility of a clinical implication of such a system.
- Another objection is why the authors chose the reserve antibiotic (vancomycin). Vancomycin should certainly not be used for prophylactic purposes. Vancomycin is a reserve antibiotic and should be stored for proven resistant strains of bacteria. I do not agree with utilization of vancomycin prophylaxis for all patients undergoing elective arthroplasty. Please comment on this.
Minor objections:
- Above all the presentation of the study is unusual. After the introduction, the authors presented the results and then the discussion. Methodology is presented after discussion. The methodology should be presented before results. Is this the journal policy?
- Results – The authors should provide exact p-values, not only p<0.05
- Methodology – Primary and secondary outcomes of the study should be clearly stated in methodology of the manuscript.
- Except in last sentence of the abstract the authors did not provide clear conclusions of the study. Please revise.
Author Response
The authors develop a method to allow direct comparison of the antibiotic concentration profile from non-absorbable, absorbable and powder depots in an artificial draining knee model. They designed a continuous flow reactor system that was programed to mimic drainage rates following revision and was similar in size to a knee- supporting a human sized PMMA bone cement spacer, absorbable calcium sulfate beads and antibiotic powder administered as a bolus. The reactor system mimicked post-surgical drainage following knee arthroplasty revision surgery to account for fluid exchange. They concluded that the combination of a vancomycin bolus, PMMA and CSB may present an optimal combination for killing bacteria entering the surgical site and provide long term protection against subsequent biofilm formation.
Major objections:
- Although the manuscript is well written it is not easy to follow. There is too much text, some parts could be more easily structured. Introduction should be shortened. Results should be structured in a Table.
Response: Page 1 lines 51-53 have been removed from the introduction. However, it was not clear which parts of the results would be better presented as a table. The text describing the results has been shortened through the removal of various phrases throughout the section. We believe figures 1-5 are best shown in graphical form as readers are able to easily see the antibiotic concentration kinetics for the different application methods over time. While Figure 5 could be presented as a table, we believe graphical format allows the reader to rapidly evaluate the differences and trends in the data. If the editor is in agreement with the reviewer that the results section is too lengthy, we could remove the zone of inhibition data shown in Figure 6 to the supplemental information.
- From a surgical point of view the most important question is the possibility of the clinical implication of the examined system. The conditions created by the authors in vitro are not realistic and are questionable in real life. In their model there is no soft tissue, there is no real absorption and later release. We do not know which amount of antibiotic should be absorbed by the tissues. Also there is no immune response. All of that should play a role in vivo. I am concerned whether there is any possibility of a clinical implication of such a system.
Response: We agree that our model, as all models oversimplifies the real human system at the expense of providing greater control. However, we believe for comparative purposes the model does have potential to incorporate the more realistic conditions of continuous fluid exchange, the rates of which changes over time as seen clinically, as opposed to conventional elution models in which infinite sink conditions are achieved by periodic full or partial exchanges. Whether our model has application for clinical prediction remains to be seen and will require verification of the predicted efficacy of one release system over another through anecdotal observation or animal studies of PJI We have now discussed this limitation on page 8 lines 869-972.
- Another objection is why the authors chose the reserve antibiotic (vancomycin). Vancomycin should certainly not be used for prophylactic purposes. Vancomycin is a reserve antibiotic and should be stored for proven resistant strains of bacteria. I do not agree with utilization of vancomycin prophylaxis for all patients undergoing elective arthroplasty. Please comment on this.
Minor objections:
- Above all the presentation of the study is unusual. After the introduction, the authors presented the results and then the discussion. Methodology is presented after discussion. The methodology should be presented before results. Is this the journal policy?
Response: The format of the manuscript was found on the author instructions page under manuscript preparation for this journal. The file format used was directly downloaded from the Antibiotics website with the formatting preset.
- Results – The authors should provide exact p-values, not only p<0.05
Response: Exact p-values have been added throughout the manuscript. There are places in the manuscript text where multiple measurement were made over periods of time in which we have used ‘P<0.05 at all timepoints’ rather than including each exact p-value.
- Methodology – Primary and secondary outcomes of the study should be clearly stated in methodology of the manuscript.
Response: The primary outcome measure is to determine the AUC for the different application methods with the hypothesis that the CSBv+tplusSpacerv+t will produce the greatest AUC dosage and have now included in the introduction on page 2 lines 201-203 and on page 11 line 1033-1034 in the methodology. While including the primary outcome to our manuscript we realize we had not included a hypothesis and have also added this on page 2 lines 198-201. We have also added the use of AUC in the abstract.
- Except in last sentence of the abstract the authors did not provide clear conclusions of the study. Please revise.
Response: We have now added a conclusions section at the end of the discussion (page 9 lines 930-939) and also expand the conclusion in the abstract to recognize our model does not assess potential cytotoxicity.
Other revisions
In addition to revisions based on reviewers comments we have also found a number of areas where grammar has been improved and have made these edits. For consistency we used hr or hrs instead of hour or hours and accepted these changes to reduce clutter in the track changes version.
We also found a mistake in our calculation of the t-test for our area under the curve data (Fig. 5) and have replaced with an updated version. Although the magnitude of the bars are exactly the same some of our P values changed. We revised the results section of the AUC accordingly. While losing some statistical significance for some of the comparisons of AUC the trends remain the same and do not effect our conclusions.
Round 2
Reviewer 2 Report
The authors improoved the manuscript and performed all required corrections.
I do not see an answer and comment in the manuscript in regards to one of my queries:
- Another objection is why the authors chose the reserve antibiotic (vancomycin). Vancomycin should certainly not be used for prophylactic purposes. Vancomycin is a reserve antibiotic and should be stored for proven resistant strains of bacteria. I do not agree with utilization of vancomycin prophylaxis for all patients undergoing elective arthroplasty. Please comment on this.
Author Response
Response: We have now removed all references to vancomycin being used as a prophylaxis in spine and primary arthroplasty. We emphasize that the use of our model is to compare elution rates from antibiotic depots used in knee revision when there is drainage and justify the use of vancomycin and tobramycin to assess this because they are commonly used in bone cement and calcium sulfate.
We have removed the following text from the introduction: “In spine surgery surgeons may sprinkle vancomycin in the form of a bolus into the surgical site as a way to increase local concentrations of antibiotics [13,14]. Although there are concerns for prophylaxis use [15], vancomycin has been used recently in revision hip and knee arthroplasty [16]”.
We have rewritten “Antibiotic loaded PMMA, CaSO4 beads and a powdered vancomycin bolus are all used by surgeons in an attempt to prevent or treat infection, however, the concentration dynamics using these different methods is poorly understood”.
as:
“Antibiotic loaded calcium sulfate (CaSO4) hemihydrate void filler beads have also been used because they are compatible with many classes of antibiotics [10] and because they are fully absorbable promoting the release of all of the antibiotic [11–13]. More recently vancomycin powder has been directly administered intraoperatively in revision hip and knee arthroplasty [14]”.
In the discussion we have replaced “Intraoperative application of vancomycin has been used as a prophylaxis in spine surgery, but as of yet the effectiveness in reducing infections are inconclusive [28,29]. More recently, a similar approach has been used in primary knee and hip arthroplasty with the intraarticular placement of vancomycin powder [30]. Otte et al. reported intrawound placement of vancomycin reduced infection in primary and revision hip and knee arthroplasties [16]. However other studies have shown that vancomycin powder had no effect on preventing infection [31,32]. It is possible that this lack of efficacy may be due to rapid washout of vancomycin, as would be predicted from our study. Furthermore, concerns are raised with the finding that a majority of patients are being underdosed leading to the recommendation that vancomycin prophylaxis for patients undergoing elective arthroplasty should be limited and carefully considered [15]”.
With “Recently, intraarticular placement of vancomycin powder has been used in primary knee and hip arthroplasty [26]. Otte et al. reported intrawound placement of vancomycin reduced infection [14] however other studies have shown that vancomycin powder had no effect on preventing infection [27,28]”.